# Direct Preference Optimization With Unobserved Preference Heterogeneity

## Abstract

RLHF has emerged as a pivotal step in aligning language models with human objectives and values. It typically involves learning a reward model from human preference data and then using reinforcement learning to update the generative model accordingly. Conversely, Direct Preference Optimization (DPO) directly optimizes the generative model with preference data, skipping reinforcement learning. However, both RLHF and DPO assume uniform preferences, overlooking the reality of diverse human annotators. This paper presents a new method to align generative models with varied human preferences. We propose an Expectation-Maximization adaptation to DPO, generating a mixture of models based on latent preference types of the annotators. We then introduce a min-max regret ensemble learning model to produce a single generative method to minimize worst-case regret among annotator subgroups with similar latent factors. Our algorithms leverage the simplicity of DPO while accommodating diverse preferences. Experimental results validate the effectiveness of our approach in producing equitable generative policies.

## 1 Introduction

Reinforcement Learning from Human Feedback (RLHF) has emerged as one of the leading methods to align Language Models (LMs) to human preferences Ouyang et al. (2022); Stiennon et al. (2020); Wang et al. (2023b). RLHF focuses on learning a single reward model from human preference data and uses that to fine-tune and align the LM. To sidestep potentially expensive reinforcement learning, Direct Preference Optimization (DPO) Rafailov et al. (2024b) is an alignment method that optimizes the LM policy directly using the preference data. However, DPO implicitly uses the same reward model as RLHF to train the LM. This reward model reflects the majority opinion of the preference data annotators and caters to that majority. If the annotator population is not representative of the general population, then this comes at the cost of neglecting groups underrepresented in the annotators, leading to misrepresentation of preferences. On the other hand, if the annotator population is representative, then opinions of minority groups in the general population are shunned, causing bias and discrimination.

Most papers that try to deal with this issue learn a reward model and then use a standard RL framework such as PPO to align the LM. However, DPO has several advantages over RLHF, eliminating the need for a reward model and leading to a more stable pipeline. Zhou et al. (2023) utilizes these benefits by developing an algorithm that directly optimizes policy by implicitly learning a multi-objective reward model. However, methods that rely on a multi-dimensional reward model Wang et al. (2024b); Zhou et al. (2023) implicitly or explicitly have two main drawbacks. First, these methods typically require annotators to rate data on a multi-dimensional scale, with each dimension corresponding to a different objective like safety or accuracy. This data is both more costly and harder to obtain compared to binary preference data Casper et al. (2023). Second, the different rating objectives must be determined ahead of the data collection stage. This can be a difficult task as there are many latent factors that might affect the preferences of annotators Siththaranjan et al. (2023), which can be difficult to discern. For example, if we collect ratings based on helpfulness and harmfulness similar to Bai et al. (2022), these rankings might not fully explain some preference decisions made because of cultural, political, or geographical inclinations.

We propose a pipeline of two algorithms to sidestep the need for RLHF for a heterogeneous population, allowing us to cater to diverse preferences without the need for reinforcement learning, letting us reap the added benefits of DPO. In particular, we propose Expectation Maximization Direct Preference Optimization (EM-DPO) and MinMax Direct Preference Optimization (MinMax-DPO). EM-DPO uses an EM algorithm Dempster et al. (1977) to simultaneously learn the distribution of user preference types as well as policies for each type. Note that, if we already knew the group each user belonged to, we could simply train an optimal DPO policy on each group separately. Since we do not, we think of our data as being generated by latent mixture model, where for each user we first draw a latent preference type and then draw a set of annotation data based on the preference type. We show that one can combine ideas from DPO with the EM algorithm for learning mixture models and directly learn a distribution of latent types, as well as a regularized optimal policy for each type. MinMax-DPO then takes these optimal policies and learns one model to best serve the needs of the population. Figure 1 shows the proposed pipeline.

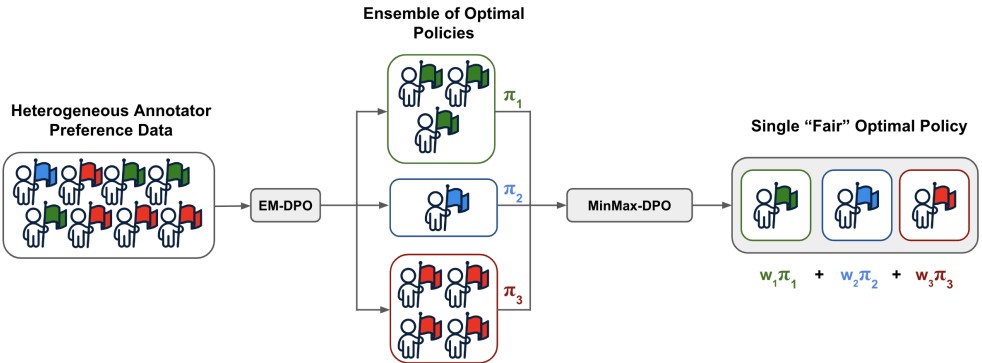

Figure 1: Proposed pipeline to find the optimal policy. Step 1: We gather binary preference data from heterogeneous annotators. Step 2: We run an expectation-maximization algorithm EM-DPO to soft assign annotators to clusters and to find an ensemble of optimal policies. Step 3: We run a regret-based algorithm Max-Min DPO to learn a linear combination of the optimal policies that is equitable.

## 2 RELATED LITERATURE

**RLHF With Diverse Preferences:** One of the chief issues in RLHF is that of diverse populations; different annotators could have very different preferences Dumoulin et al. (2023). Several studies have tried to solve the diverse population problem by learning more expressive reward functions and then using them to perform RLHF. For example, Rame et al. (2024); Jang et al. (2023); Chakraborty et al. (2024) maintains and learns several reward models at once. Similarly, Wang et al. (2024b) learns a multi-dimensional reward model where each dimension provides rewards based on a different objective such as safety or usefulness. Yang et al. (2024) proposes a policy-agnostic method to perform multi-objective LLM alighment. Alternatively, Siththaranjan et al. (2023); Li et al. (2024) learns a distribution over fixed reward models. Finally, these reward models are combined using various strategies Bakker et al. (2022); Jang et al. (2023); Rame et al. (2024) to get a final reward model which is then used to perform RLHF. Chakraborty et al. (2024) also learns multiple reward models, but performs RL by maximizing the minimum reward thereby ensuring that the final model is fair. The paper draws on elements of social choice theory, which Conitzer et al. (2024) argues is an effective path forward for RLHF research in general, specifically regarding issues with aggregating preferences. Dai & Fleisig (2024) outlines a correspondence between the key principles and desiderata of social choice into the RLHF context.

In an orthogonal approach, Zhong et al. (2024) utilizes meta-learning to learn diverse preferences. In general, trying to do RLHF with many reward models becomes expensive, making extending DPO

Rafailov et al. (2024b) an attractive alternative. Swamy et al. (2024) proposes SPO to sidestep reinforcement learning using the concept of a minimax winner from social choice theory, but only in the case of homogeneous preferences. In concurrent work, Park et al. (2024) proposes a personalized RLHF algorithm which learns clustered policies via a hard Expectation Maximization algorithm using DPO. We instead propose a soft-clustering algorithm, which enjoys stronger theoretical guarantees Dempster et al. (1977). Park et al. (2024) also proposes an algorithm to aggregate estimated reward functions for a heterogeneous population. Ramé et al. (2024) also deals with the idea of aggregating reward models to increase robustness. We instead propose a complete pipeline to learn one equitable policy for a heterogeneous population without appealing to reward model estimation at all.

**DPO Generalizations:** Since DPO's inception Rafailov et al. (2024b), there has been a growing line of literature on its generalizations, some of which we highlight here. Le et al. (2024) generalizes DPO to the case of multiple SFT models, while Zhou et al. (2024) generalizes to multiple objectives. Zeng et al. (2024); Rafailov et al. (2024a) work on extending DPO to work at the token level. Wang et al. (2023a) extends DPO to work with other types of divergence terms, while Wu et al. (2024) relates DPO to DRO in order to robustify it. Badrinath et al. (2024) augments DPO with a computable advantage function to create a hybrid between DPO and RLHF.

Additional work that more generally relates to the fields of reward modeling and preference-based reinforcement learning can be found in Appendix A.

## 3 BACKGROUND

In this section, we discuss traditional alignment methods that assume uniform preference among the whole population, namely RLHF Ziegler et al. (2019); Stiennon et al. (2020); Ouyang et al. (2022) and DPO Rafailov et al. (2024b).

**Reinforcement Learning from Human Feedback (RLHF)** The RLHF pipeline has two inputs. The first is an LM $\pi_{\text{SFT}}$ that is pre-trained on internet-scale data and then fine-tuned with supervised learning. The second input is a static annotator preference dataset. To collect this data, first pairs of responses $(y_1, y_2)$ are generated from $\pi_{\text{SFT}}(\cdot|x)$ where $x$ is a given prompt. Human annotators then choose the best response between the two - in what follows, let $y_1$ denote the winning response and $y_2$ denote the losing response. Also, let $\mathcal{H}$ be the population of all human annotators and $h \in \mathcal{H}$ be the random variable that represents a single human annotator.

In the first step of RLHF, a reward model $r_\phi(x, y)$ is fit using the preference data. This is done by minimizing the following log-likelihood loss:

$$\mathcal{L}(r_\phi; \mathcal{D}) = -\mathbb{E}_{(x,y_1,y_2,h)\sim\mathcal{D}}[p(y_1 \succ y_2|x)] \tag{1}$$

To simplify this objective, assume that the relation between preference data and rewards follows the Bradley-Terry-Luce model Bradley & Terry (1952). Let $r^*(x, y)$ represent the true rewards for all annotators. Then, according to the BTL model, the probability that an annotator prefers one response over the other is given by:

$$p(y_1 \succ y_2|x) = \frac{\exp(r^*(x, y_1))}{\exp(r^*(x, y_1)) + \exp(r^*(x, y_2))} = \sigma(r^*(x, y_1) - r^*(x, y_2)), \tag{2}$$

so the log-likelihood loss to minimize is equivalent to

$$\mathcal{L}(r_\phi; \mathcal{D}) = -\mathbb{E}_{(x,y_1,y_2,h)\sim\mathcal{D}}[\sigma(r_\phi(x, y_1) - r_\phi(x, y_2))] \tag{3}$$

The second and final step is fine-tuning with reinforcement learning (RL) using the learned reward model $r_\phi(x, y)$. More specifically, the Proximal Policy Optimization (PPO) Schulman et al. (2017) is used in training the LM. The PPO algorithm optimizes the following objective:

$$\pi_\theta^* = \arg\max_{\pi_\theta} \mathbb{E}_{x\sim\mathcal{D}, y\sim\pi_\theta(y|x)}[r_\phi(y, x)] - \beta_{\text{KL}}[\pi_\theta(y|x)||\pi_{\text{SFT}}(y|x)] \tag{4}$$

**Direct Preference Optimization (DPO)** DPO optimizes the same objective as PPO as given in 4 but bypasses learning the reward model by directly optimizing with the preference data by combining

2 and 4. This results in a pipeline that is not only significantly simpler, but also exhibits greater stability Rafailov et al. (2024b).

DPO minimizes the following log-likelihood loss directly using preference data to obtain $\pi_\theta^*$:

$$
\mathcal{L}(\pi_\theta; \pi_{\text{SFT}}, \mathcal{D}) = -\mathbb{E}_{(x,y_w,y_l)\sim\mathcal{D}} \left[ \log \sigma \left( \beta \log \frac{\pi_\theta(y_w|x)}{\pi_{\text{SFT}}(y_w|x)} - \beta \log \frac{\pi_\theta(y_l|x)}{\pi_{\text{SFT}}(y_l|x)} \right) \right]
$$
$$
\pi_\theta^* = \arg\min_{\pi_\theta} \mathcal{L}(\pi_\theta; \pi_{\text{SFT}}, \mathcal{D})
$$
(5)

Both RLHF and DPO assume that preferences are uniform across the population and implicitly or explicitly learn a single reward model. However, this is not the case in the real world as humans have diverse preferences and values. Moreover, RLHF and DPO prioritize the majority opinion of the annotator population. This could lead to misalignment if the annotation population is not representative of the general population. Traditional methods like RLHF and DPO can therefore lead to bias and discrimination towards the minority subgroups among the annotator population. We propose a new algorithm, MinMax-DPO, that learns an equitable and fair optimal policy directly from binary preference data to bridge this gap.

## 4 EM-DPO: PROBABILISTIC DIRECT PREFERENCE OPTIMIZATION ALGORITHM

The Expectation-Maximization Algorithm Dempster et al. (1977); Moon (1996) deals with settings with mixture data. Data are produced by first drawing a set of latent factors $Z$ and then drawing a set of observed variables $V \mid Z$. The parameters of the likelihood determine both the distribution of the latent factors $p(Z; \theta)$ as well as the conditional likelihood $p(V \mid Z; \theta)$. At step $t$ of the algorithm, we have a current candidate parameter vector $\theta_t$ and calculate $\theta_{t+1}$ as follows:

$$
\theta_{t+1} = \arg\max_\theta Q(\theta \mid \theta_t) := \mathbb{E}_{Z\sim p(\cdot|V,\theta_t)} \left[ \log(p(V, Z \mid \theta)) \right]
$$
(6)

In our setting, the latent factors $Z = (Z_1, \ldots, Z_n)$ correspond to the unobserved heterogeneity types $Z_i$ of an annotator $i \in [n]$ and $V = (V_1, \ldots, V_m)$ correspond to the chosen preferences $y_1^{ij} \succeq y_2^{ij}$ for each of the prompts $X_{ij}$ assigned to the annotator. We assume for simplicity that each annotator is assigned $m$ prompts and we let $V_{ij} = (X_{ij}, y_1^{ij} \succeq y_2^{ij})$, where $X_{ij}$ is the prompt and $y_1^{ij} \succeq y_2^{ij}$ is the preference for that prompt. Our parameters $\theta$ are $(\phi, \rho, \eta)$, where $\phi$ are the parameters for the group-wise policies, $\eta$ the latent distribution of user types and $\rho$ are parameters that determine the distribution of prompts $X_{ij}$.

With some calculation, we find that a parameterization of the policy $\pi_{\phi,z}$ implies a parameterization of the likelihood (see Appendix B):

$$
p(V_i \mid Z_i; \theta) = \prod_{j=1}^m \sigma_\phi(Z_i, V_{ij}) \, p(X_{ij} \mid Z_i; \rho)
$$
(7)

where the function $\sigma_\phi$ is similar to the parameterization introduced in DPO:

$$
\sigma_\phi(z, x, y_1, y_2) := \sigma \left( \beta \log \frac{\pi_{\phi,z}(y_1|x)}{\pi_{\text{SFT}}(y_1|x)} - \beta \log \frac{\pi_{\phi,z}(y_2|x)}{\pi_{\text{SFT}}(y_2|x)} \right)
$$
(8)

Note that the latent factors take values in a set of $K$ discrete values $\{z_1, \ldots, z_K\}$. In this case, we can assume a fully non-parametric likelihood $p(Z; \theta)$, where $\eta = p(z_k; \theta) \in \Delta(K)$, the $K$-dimensional simplex. Subsequently, we can decompose the criterion as:

$$
Q(\theta \mid \theta_t) = \mathbb{E}_{Z\sim p(\cdot|V,\theta_t)} \left[ \sum_{i=1}^n \log(p(V_i, Z_i \mid \theta)) \right] = \mathbb{E}_{Z\sim p(\cdot|V,\theta_t)} \left[ \sum_{i=1}^n \log(p(V_i \mid Z_i; \theta)) + \log(p(Z_i; \theta)) \right]
$$
(9)

For further simplification, we note that $p(Z_i; \theta) = \sum_{k=1}^{K} \eta_k \mathbf{1}\{Z_i = z_k\}$ to get that

$$Q(\theta \mid \theta_t) = \mathbb{E}_{Z \sim p(\cdot \mid V, \theta_t)} \left[ \sum_{i=1}^{n} \log(p(V_i \mid Z_i; \theta)) + \log \left( \sum_{k=1}^{K} \eta_k \mathbf{1}\{Z_i = z_i\} \right) \right] \quad (10)$$

Assuming that $p(V_i \mid Z_i; \theta)$ does not depend on the vector $\eta$, so that $p(V_i \mid Z_i; \theta) = p(V_i \mid Z_i; \phi, \rho)$, the original criterion decomposes into two separate optimization problems:

$$\eta_{t+1} = \underset{\eta}{\arg\max}\, \mathbb{E}_{Z \sim p(\cdot \mid V, \theta_t)} \left[ \sum_{i=1}^{n} \log \left( \sum_{k=1}^{K} \eta_k \mathbf{1}\{Z_i = z_k\} \right) \right]$$

$$\phi_{t+1} = \underset{\phi, \rho}{\arg\max}\, \mathbb{E}_{Z \sim p(\cdot \mid V, \theta_t)} \left[ \sum_{i=1}^{n} \log(p(V_i \mid Z_i; \phi, \rho)) \right] \quad (11)$$

For the $E$-step, we must characterize the posterior distribution of the latent factors. Under the assumption that the contexts are un-correlated with the unobserved preference types, which is natural in the context of LLM fine-tuning, since contexts are randomly assigned to annotators, we can derive that (see Appendix C):

$$p(z_k \mid V_i; \theta) = \frac{\eta_k \prod_{j=1}^{m} \sigma_\phi(z_k, V_{ij})}{\sum_{\ell=1}^{K} \eta_\ell \prod_{j=1}^{m} \sigma_\phi(z_\ell, V_{ij})} \quad (12)$$

For the $M$-step, we must solve the two optimization problems given above. The solution for $\eta$ can be derived in closed form, while the solution for $\phi$ is independent of the term $p(X_{ij} \mid Z_i; \rho)$:

$$\eta_{k,t+1} = \frac{1}{n} \sum_{i=1}^{n} p(z_k \mid V_i; \theta_t) \quad (13)$$

$$\phi_{t+1} = \underset{\phi}{\arg\max} \sum_{i=1}^{n} \mathbb{E}_{Z_i \sim p(\cdot \mid V_i; \theta_t)} \left[ \sum_{j=1}^{m} \log(\sigma_\phi(Z_i, V_{ij})) \right] \quad (14)$$

A full derivation is in Appendix D. This gives rise to the following EM algorithm:

---

**Algorithm 1** EM-DPO: Expectation-Maximization Direct Preference Optimization

---

[1] **Input:** Preference data $\mathcal{D}$ indexed for all human annotators $\mathcal{I}$ and containing $m_i$ demonstrations for each human annotator $i$. **Input:** pre-trained group-wise models $\pi_{\phi_0, z}; \forall z \in \{z_1, \ldots, z_k\}$. Initialize $\eta_0 = (1/K, \ldots, 1/K)$ $t$ in $\{0, \ldots, T\}$ **E.** Calculate posterior $p(z_k \mid V_i; \theta_t)$ for each annotator $i \in \mathcal{I}$:

$$\gamma_{i,k} = \frac{\eta_{k,t} \prod_{j=1}^{m_i} \sigma_{\phi_t}(z_k, V_{ij})}{\sum_{\ell=1}^{K} \eta_{\ell,t} \prod_{j=1}^{m_i} \sigma_{\phi_t}(z_\ell, V_{ij})}$$

**M.** Update parameters $\phi, \eta$:

$$\eta_{k,t+1} = \frac{\sum_{i \in \mathcal{I}} \gamma_{i,k}}{|\mathcal{H}|}$$

$$\phi_{t+1} = \arg\max_\phi \sum_{i \in \mathcal{I}} \sum_{k=1}^{K} \gamma_{i,k} \sum_{j=1}^{m_i} \log(\sigma_\phi(z_k, V_{ij}))$$

**Return:** Policies $\{\pi_{\phi_t, z} : z \in \{z_1, \ldots, z_k\}\}$ and posterior preference weights $\{\gamma_{i,k} : i \in \mathcal{I}\}$.

---

Note that if we do not share parameters across the policies for each preference type $z$, i.e. we have separate parameters $\phi_z$ for each $z \in \{z_1, \ldots, z_K\}$, then the optimization in the final step of EM-DPO also decomposes into separate policy optimization problems for each preference type:

$$\phi_{z_k, t+1} = \underset{\phi_{z_k}}{\arg\min} \sum_{i \in \mathcal{I}} \sum_{j=1}^{m_i} \gamma_{i,k} \log(\sigma_\phi(z_k, V_{ij})) \quad (15)$$

Note that the latter is simply a weighted DPO problem, where each demonstration $(h, j)$, which corresponds to the $j$-th demonstrations from annotator $i$, is assigned weight $\gamma_{i,k}$ when optimizing the policy parameters for preference type $z_k$. Alternatively, for multi-tasking purposes, some parameters can be shared parameters across policies for each preference type, in which case the final optimization problem should be solved simultaneously via stochastic gradient descent over the joint parameters $\phi$.

# 5 MINMAX-DPO: DIRECT OPTIMIZATION FOR MIN-MAX REGRET ENSEMBLE

## 5.1 MINMAX REGRET OBJECTIVE

So far, we have shown how to calculate a separate policy that optimizes for each preference population $z$. Our ultimate goal is to output a single policy. Hence, we need to trade-off optimizing for the preferences of different groups and find a policy that strikes a good balance.

In that respect, to equitably cater to all $K$ sub-populations, we focus on identifying a policy that minimizes the worst-case regret among the sub-populations. To avoid having to retrain a new policy, we will restrict ourselves to selecting an ensemble among the already trained policies. As such, we define the ensemble space of policies as:

$$\Pi = \left\{ \sum_{k=1}^{K} w_k \pi_{\phi, z_k} : w \in \Delta(K) \right\} \tag{16}$$

If we had access to the reward functions $r_z^*(y, x)$, then for any policy $\pi$, the expected reward that population $z$ receives would be:

$$R_z(\pi) = \mathbb{E}_{x \sim \mathcal{D}, y \sim \pi(\cdot|x)} \left[ r_z^*(y, x) \right] \tag{17}$$

Note that if we were to solely focus on population $z$, we would be optimizing the expected reward objective above, regularized so as not to deviate from the reference policy. This would yield policy $\pi_z^* = \pi_{\phi^*, z}$, where $\phi_*$ are the policy parameters we calculated based on the EM-DPO algorithm.

Our goal is to find an ensemble policy $\pi$ such that no population $z$ has very large regret towards choosing their population-preferred policy $\pi_z^*$. Our minimax regret optimization problem can be simply stated as:

$$\pi_* = \arg \min_{\pi \in \Pi} \max_{k=1}^{K} \left[ R_{z_k}(\pi_{z_k}^*) - R_z z_k \pi \right]^+ \tag{18}$$

where $[x]^+ = \max\{x, 0\}$. Note that we only consider the positive part of the regret.

**Why min-max regret?** Max-min reward is another fairness criterion that can be applied to the RLHF problem to ensure equity, as discussed in Chakraborty et al. (2024). However, this criterion has two major drawbacks. Firstly, the reward model is not uniquely identifiable from preference data. Two reward models $r(x, y)$ and $r'(x, y)$ are equivalent if $r(x, y) - r'(x, y) = f(x)$ Rafailov et al. (2024b). Therefore, directly maximizing the minimum reward is ineffective due to this scaling. We could fix this by standardizing the reward model to set the minimum reward to zero - if $r(x, y)$ is the recovered reward function, we can use $r'(x, y) = r(x, y) - \min_y r(x, y)$, which is an equivalent reward model. Even then, there is another issue with the max-min reward criterion: The max-min reward focuses on improving rewards for users with the lowest reward, while the min-max regret function targets users with the highest regrets. These groups differ when users with low rewards also have low regrets. As an example, consider a setting with fixed context and three responses. If two users have reward vectors [0, 0.01, 0.02] and [0, 10, 1] respectively, then the max-min reward objective will choose response 3 to maximize user 2's reward. However, user 1 is nearly indifferent between the three choices 2, whereas user 2 strongly prefers option 2. Therefore, it is more ideal to choose option 2, which the min-max regret criteria chooses.

## 5.2 REGRET DYNAMICS

We now show that the min-max regret objective can also be optimized over, without access to the explicit reward functions, but solely based on the policies we have already trained. We can rewrite

our objective as (see Appendix E):

$$\min_{w \in \Delta(K)} \max_{z \in \{z_0, z_1, \ldots, z_K\}} \sum_{k=1}^{K} w_k \cdot (\mathcal{L}_{z,z} - \mathcal{L}_{z,z_k}),$$

(19)

where

$$\mathcal{L}_{z,z'} := \mathbb{E}_{x \sim \mathcal{D}, y \sim \pi_{z'}^*(\cdot|x)} \left[ \log \left( \frac{\pi_z^*(y|x)}{\pi_{\text{SFT}}(y|x)} \right) \right].$$

(20)

Letting $\mathcal{R}$ denote the $(K+1) \times K$ matrix whose $(k, k')$ entry (for $0 \le k \le K, 1 \le k' \le K$) corresponds to $\mathcal{R}_{k,k'} := \mathcal{L}_{z_k,z_k} - \mathcal{L}_{z_k,z_{k'}}$, we can re-write the above objective as:

$$\min_{w \in \Delta(K)} \max_{p \in \Delta(K+1)} p^\top \mathcal{R} w$$

(21)

This is simply a finite action zero-sum game, where the minimizing player has $K$ actions and the maximizing player has $K + 1$ actions. A large variety of methods can be utilized to calculate an equilibrium of this zero-sum game and hence identify the minimax regret optimal mixture weights $w_*$. For instance, we can employ optimistic Hedge vs. optimistic Hedge dynamics, which are known to achieve fast convergence rates in such finite action zero-sum games Rakhlin & Sridharan (2013) and then use the average of the solutions over the iterates of training, as described in Algorithm 2.

---

**Algorithm 2** MinMax-DPO: Direct Optimization for Min-Max Regret Ensemble

[1] **Input:** Distribution $\mathcal{D}$ of contexts $x$. **Input:** Population-specific optimal policies $\pi_z^* \equiv \pi_{\phi_*,z}$ returned from EM-DPO **Input:** Number of iterations $T$ and a sufficiently small, albeit constant, independent of $T$, step-size $\eta$ Calculate discrepancies for $z, z' \in \{z_1, \ldots, z_k\}$:

$$\mathcal{L}_{z,z'} := \mathbb{E}_{x \sim \mathcal{D}, y \sim \pi_{z'}^*(\cdot|x)} \left[ \log \left( \frac{\pi_z^*(y|x)}{\pi_{\text{SFT}}(y|x)} \right) \right]$$

with the convention that $\mathcal{L}_{z_0,z_0} = \mathcal{L}_{z_0,z_k} = 0$ Calculate $(K+1) \times K$ regret matrix $\mathcal{R}$, whose $k \in \{0, \ldots, K\}$ and $k' \in \{1, \ldots, K\}$ entry is:

$$\mathcal{R}_{k,k'} := \mathcal{L}_{z_k,z_k} - \mathcal{L}_{z_k,z_k'},$$

Initialize $w_0 = (1/K, \ldots, 1/K)$ and $p_0 = (1/(K+1), \ldots, 1/(K+1))$ $t$ in $\{0, \ldots, T\}$

$$w_t \propto w_{t-1} \exp \left\{ -\eta \cdot \left( 2\mathcal{R}^\top p_{t-1} - \mathcal{R}^\top p_{t-2} \right) \right\}$$

$$p_t \propto p_{t-1} \exp \left\{ \eta \cdot (2\mathcal{R} w_{t-1} - \mathcal{R} w_{t-2}) \right\}$$

**Return:** Policy $\pi_* = \sum_{k=1}^{K} w_k^* \pi_{\phi_*,z_k}$, where $w^* = \frac{1}{T} \sum_{t=1}^{T} w_{k,t}$

---

The solution $\pi_*$ returned by Algorithm 2 consistutes a $O(\log(K) \log(T) T^{-1})$-approximate solution to the min-max regret problem (a direct consequence of the results in Rakhlin & Sridharan (2013)). This completes our overall direct preference optimization procedure with unobserved heterogeneous preferences.

One can also optimize a new policy $\pi$ that does not correspond to an ensemble of the base policies $\pi_{\phi_*,z}$ by solving the saddle point problem:

$$\min_\pi \max_z L_z(\pi) R_z(\pi) - R_z(\pi_z^*)$$

(22)

which has already been shown, can be expressed as a function of $\pi_z^*$ and $\pi_{\text{SFT}}$. This saddle point can be solved by policy-gradient vs multiplicative weight dynamics, or for faster convergence via optimistic policy gradient descent vs optimistic mulitplicative weight dynamics:

$$\phi_{t+1} = \phi_t - 2\nabla_\phi \sum_z p_{t,z} L_z(\pi_{\phi_t}) + \nabla_\phi \sum_z p_{t-1,z} L_z(\pi_{\phi_t-1})$$

(23)

$$p_{t+1,z} \propto p_{t,z} \exp\{\eta \cdot (2L_z(\pi_{\phi_t}) - L_z(\pi_{\phi_t-1}))\}$$

(24)

## 6 EXPERIMENTS

### 6.1 MULTI-ARMED BANDIT EXPERIMENT

#### 6.1.1 SETTINGS

We can draw parallels between the offline contextual bandit problem and the problem of learning from human preferences Azar et al. (2024). In this setting, the context represents the prompt and the bandit arms represent the possible responses for the given context. For our experiment, we consider a simplified case with three arms.

First, we generate 200 annotators drawn randomly from three sub-populations, with 60% of the population coming from the first sub-population, 30% from the second, and 10% from the third. The preferences of annotators within each sub-population is homogeneous and therefore, each sub-population is associated with a single reward model. For our experiment, we model the sub-population reward model using a linear function, similarly to linear contextual bandits Dimakopoulou et al. (2019):

$$r_z^*(y, x) = x^T \theta_z(y) + \mathcal{N}(0, \sigma) \tag{25}$$

where $\theta_z(y)$ is the model parameters corresponding to the latent variable $z$ for arm $y$, $x$ is the context, and $\mathcal{N}(0, \sigma)$ represents noise with 0 mean and standard deviation $\sigma = 0.01$. $\theta_z(y)$ is fixed for any given sub-population with values $\theta_i(i) = [10, 10, 10]$ and $\theta_i(j) = [0, 0, 0], j \neq i$ for sub-population $i$. We generate 10 preference data pairs per annotator. For each data point, first we draw a context vector $x$ uniformly randomly from the hypercube $[0, 1]^3$. Then, a pair of responses is generated from a uniformly random reference policy $\pi_{\text{SFT}} = (1/3, 1/3, 1/3)$. The annotator then chooses a response $y_w \succ y_l$ based on their reward model $r_z^*(y, x)$. We implement EM-DPO and MinMax-DPO for this data. Appendix F shows hyperparameters for the experiment.

#### 6.1.2 RESULTS

We run standard DPO and MinMax-DPO on this experimental setup and calculate the average regret per user group. The results of this are shown in Figure 2.

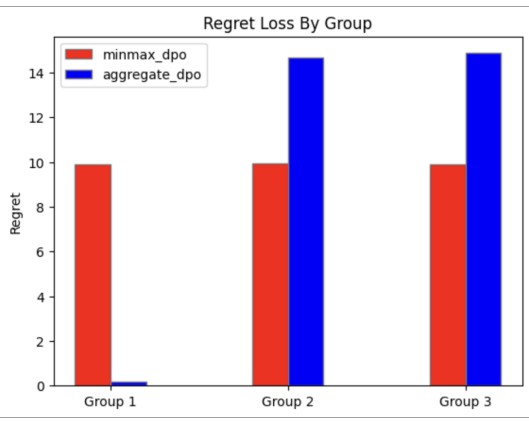

Figure 2: DPO vs. MinMax-DPO Regret Plot

We can see that training DPO over the whole population leads to the policy completely optimizing for the first user group's preference (i.e., majority opinion), leading to maximal regret for the other two groups. However, MinMax-DPO achieves the social optimum and respects the preferences of all three groups, as shown by the equal regret among all three groups. Figure 3 shows the convergence of the learned weights in the MinMax-DPO algorithm; we see relatively quick convergence to the optimal weights, which are close to uniform. This is expected as all three sub-groups have perfectly contradicting opinions because they each prefer a different response.

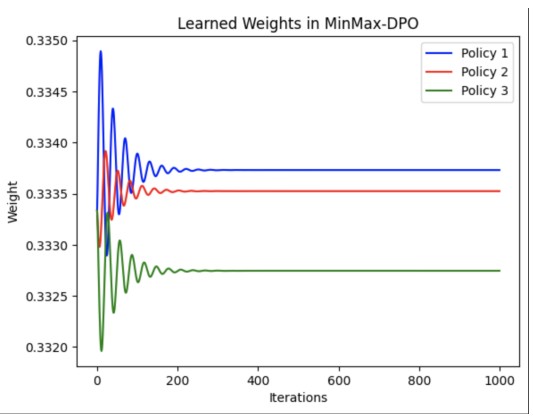

Figure 3: Convergence of learned weights in MinMax-DPO.

## 6.2 LLM EXPERIMENT ON THE IMDB DATASET

### 6.2.1 SETTINGS

We conduct this experiment on GPT-2 Large, a 774M parameter version of GPT-2 Radford et al. (2019). We use a synthetically generated dataset using the IMDb reviews Maas et al. (2011) as the preference dataset. This dataset contains 20,000 preference data points that we assign equally (1,000 preferences per person) to 20 users from 2 different user sub-groups. The first sub-group (Group 1) is majority and comprises of 15 users; this group always prefers the more grammatically correct response. The second sub-group (Group 2) is minority, containing 5 users; they always prefer the shorter response. Appendix F contains more details on how the data is generated.

### 6.2.2 ALGORITHMS

We showcase metrics for 5 policies: (1) `MinMax-DPO Policy`: The ensemble of `Group 1 Policy` and `Group 1 Policy` learned during Algorithm 2. (2) `Naive DPO Policy`: We get this policy by simply running DPO on the full preference dataset without learning any sub-populations. (3) `Group 1 Policy` and `Group 2 Policy`: The optimal policies for each of the sub-groups learned during the EM-DPO algorithm. (4) `Cluster-DPO Policy`: First, we perform $k$-means clustering on the chosen responses. Then we train DPO on each of these preference clusters. Finally, we run Algorithm 2 on these policies to get this ensemble policy.

### 6.2.3 RESULTS

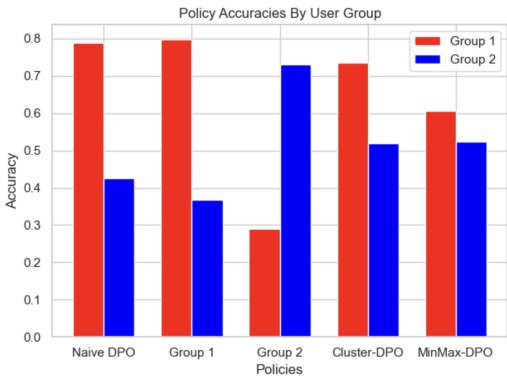

Figure 4: The accuracy on an evaluation dataset of the five policies: Naive DPO, Group 1, Group 2, Cluster-DPO, and MinMax-DPO

Figure 4 shows the accuracy on an evaluation dataset of 256 data points per user group. Here, we define accuracy as the percentage of data points $(x, y_1, y_2)$, where $x$ is the prompt, $y_1$ is the chosen response, and $y_2$ is the rejected response, such that

$$\beta \frac{\pi_{\phi,z}(y_1|x)}{\pi_{\text{SFT}}(y_1|x)} > \beta \frac{\pi_{\phi,z}(y_2|x)}{\pi_{\text{SFT}}(y_2|x)},$$

or equivalently, the percentage of data points where the chosen response is given a higher "reward" than the rejected one.

As expected, Naive DPO caters to group 1, the majority group, with an accuracy of about 80%, while the minority group only has an accuracy of about 40%. The optimal policies for group 1 and group 2 have high accuracy for their respective group and low accuracy for the other, but the final ensembled MinMax-DPO policy is much more equitable, achieving 60.5% accuracy on user group 1 and 52.3% accuracy on user group 2. The Cluster-DPO policy achieves a 51.9% accuracy on user group 2, but a higher accuracy on group 1. MinMax-DPO slightly outperforms Cluster-DPO in terms of fairness as the worse off group (group 2) has slightly higher accuracy for MinMax-DPO. One of our preference metrics - length - is fairly easy to identify for a naive clustering algorithm, which could explain its superior performance for group 1. We anticipate that this method would perform worse in situations where preferences are more complex.

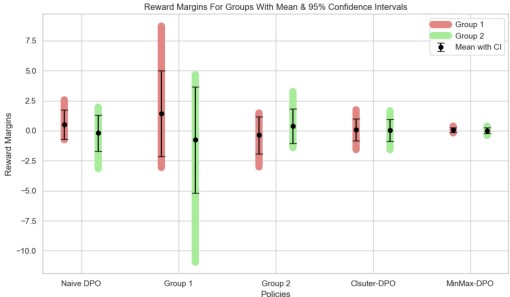

Figure 5: The reward margins on an evaluation dataset of the five policies.

Figure 5 shows the reward margins as defined as:

$$r_m(x, y_1, y_2) = \beta \frac{\pi_{\phi,z}(y_1|x)}{\pi_{\text{SFT}}(y_1|x)} - \beta \frac{\pi_{\phi,z}(y_2|x)}{\pi_{\text{SFT}}(y_2|x)}$$

on the evaluation dataset for each of the five policies. The black dots highlight the mean margin and the black bars represent 95% confidence intervals. We see that, again, the MinMax-DPO is more equitable than the naive policy with slightly positive mean margins for both user groups, whereas the naive DPO has a negative mean margin for the minority user group. Out of all policies, the worse mean margin out of the two user groups is highest for the MinMax-DPO policy, further showing equitability.

## 6.3 DISCUSSION & LIMITATIONS

We provide a robust framework to train equitable policies for a heterogeneous population with diverse preferences. By extending the DPO algorithm, we are able to sidestep reinforcement learning entirely, enjoying the added stability that DPO provides while making it more applicable to real-world situations and datasets. We demonstrate our findings on a contextual bandit experiment as well as a larger-scale LLM experiment, showing how our algorithm, MinMax-DPO, generates a far more socially equitable policy than standard DPO in diverse populations where some groups may be underrepresented.

Based on our results, we raise some limitations and directions for future work. Our derivations operate off of the assumption that contexts are uncorrelated given the preference type of the annotator; this may not be the case in the real world as, to increase accuracy of data collection, annotators may be given prompts more tuned to their skill sets. We also assume annotators report their preferences honestly, which may not be the case - this raises important questions regarding incentive compatibility.

## 7 REPRODUCIBILITY STATEMENT

We have tried to make the results in this paper as reproducible as possible. Appendix B through Appendix E contain a complete derivation of the equations required for the algorithms. Appendix F contains all the hyperparameters used to run the experiments, including the randomness seeds so that the exact figures can be replicated. Appendix F also contains further information on data generation for the LLM experiment.

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

## A  ADDITIONAL RELATED WORK

**Preference-Based Reinforcement Learning:** Reinforcement learning from preferences has been an active research area for some time, providing a way to train on tasks for which explicitly defining rewards is hard Wirth et al. (2017); Lee et al. (2021); Abdelkareem et al. (2022). In particular, Christiano et al. (2017); Ibarz et al. (2018) show that using human preferences to guide reinforcement learning (RLHF) is particularly effective on a variety of tasks, such as training robots. More recently, RLHF has become a very popular technique to fine-tune language models to do a variety of tasks such as summarization Ouyang et al. (2022); Ziegler et al. (2019); Stiennon et al. (2020); Wu et al. (2021). RLHF has also been used to align language models Bai et al. (2022); Askell et al. (2021). Casper et al. (2023) details several open problems in the field of RLHF, including those related to the feedback itself, particularly the inverse relation between richness and efficiency. Some work has been done on this problem with regards to language-based feedback in particular Fu et al. (2019); Zhou & Small (2021) as well as in more general settings Hwang et al. (2024), but specific applications to LLMs have not been fully explored.

**Challenges with Reward Modeling:** In general, human preferences can be difficult to represent using reward models Hong et al. (2022), and the validity of reward modeling itself is still somewhat debated Bowling et al. (2023); Bobu et al. (2023); Skalse & Abate (2022). Some work has also been done to take personality into account when reward modeling Lindner & El-Assady (2022); Lee et al. (2021), but this area remains open. In general, taking human irrationality into account when reward modeling (to optimize a more accurate reward function) leads to a trade-off between efficiency and accuracy Shah et al. (2019); Nguyen et al. (2017). Work has been done on inverse RL with particular models of suboptimality such as myopia Evans et al. (2016), noise Zheng et al. (2014), and risk-sensitivity Majumdar et al. (2017), but dealing with general irrationalities remains open.

The proper use and collection of data remains an issue with RLHF. Sun et al. (2024) analyzes LLM fine-tuning as a mechanism design problem where agents may have the incentive to misreport their preferences. Data can also often have issues or certain data points may not be as effective as others; Wang et al. (2024a) proposes methods to deal with incorrect or ambiguous preference pairs, while Yin et al. (2024) proposes an extension to DPO which uses contrastive learning to discern between more and less preferred responses to prompts.

## B  LIKELIHOOD PARAMETERIZATION

Note that, in our situation, the latent factors and observed variables $(Z_i, V_i)$ are independent across annotators and therefore, the likelihood and the prior factorizes across the annotators. Moreover, conditional on the latent factor, the $V_{ij}$ are independently distributed across $j$ and for each $j$ the conditional likelihood takes a logistic form, as follows:

$$p(V_i \mid Z_i; \theta) = \prod_{j=1}^{m} p(V_{ij} \mid Z_i; \theta) = \prod_{j=1}^{m} p(y_1^{ij} \succeq y_2^{ij}, X_{ij} \mid Z_i; \theta) \tag{26}$$

$$= \prod_{j=1}^{m} p(y_1^{j} \succeq y_2^{ij} \mid X_{ij}, Z_i; \theta) \, p(X_{ij} \mid Z_i; \theta) \tag{27}$$

$$= \prod_{j=1}^{m} \sigma \left( r^* \left( Z, X_{ij}, y_1^{ij} \right) - r^* \left( Z, X_{ij}, y_2^{ij} \right) \right) p(X_{ij} \mid Z_i; \theta), \tag{28}$$

where $r^*$ denotes the true reward for the annotator, as in Section 3.

The first part $\sigma \left( r^*(Z, X_j, y_1^{j}) - r^*(Z, X_j, y_2^{j}) \right)$ can also be written in closed form in terms of the policy parameters $\pi_{\phi^*, z}$ for each preference type as designated by the same observation as in Rafailov et al. (2024b):

$$\sigma(r^*(z, x, y_1) - r^*(z, x, y_2)) = \sigma \left( \beta \log \frac{\pi_{\phi^*, z}(y_1 | x)}{\pi_{\text{SFT}}(y_1 | x)} - \beta \log \frac{\pi_{\phi^*, z}(y_2 | x)}{\pi_{\text{SFT}}(y_2 | x)} \right) \tag{29}$$

where $\pi_{\phi^*,z}$ optimizes the type specific regularized objective:

$$\pi_{\phi^*,z} = \arg\max_{\pi} \mathbb{E}_{x\sim\mathcal{D},y\sim\pi(y|x)}[r^*(z,x,y)] - \beta_{\mathrm{KL}}[\pi(y|x)||\pi_{\mathrm{SFT}}(y|x)] \tag{30}$$

We will introduce the shorthand notation:

$$\sigma_\phi(z,x,y_1,y_2) := \sigma\left(\beta\log\frac{\pi_{\phi,z}(y_1|x)}{\pi_{\mathrm{SFT}}(y_1|x)} - \beta\log\frac{\pi_{\phi,z}(y_2|x)}{\pi_{\mathrm{SFT}}(y_2|x)}\right) \tag{31}$$

Thus a parameterization of the policy space $\pi_{\phi,z}$, implies a parameterization of the likelihood:

$$p(V_i \mid Z_i;\theta) = \prod_{j=1}^{m} \sigma_\phi(Z_i, V_{ij})\,p(X_{ij} \mid Z_i;\theta), \tag{32}$$

as desired.

## C $E$-STEP DERIVATION

Here, we derive the posterior distribution $p(Z \mid V;\theta) = \prod_{i=1}^{n} p(Z_i \mid V_i;\theta)$ for any given parameter $\theta$. We apply Bayes rule:

$$p(z_k \mid V_i;\theta) = \frac{p(V_i, z_k;\theta)}{p(V_i;\theta)} = \frac{p(V_i \mid z_k;\theta)\,p(z_k;\theta)}{\sum_{\ell=1}^{K} p(V_i \mid z_\ell;\theta)\,p(z_\ell;\theta)} = \frac{p(V_i \mid z_k;\phi)\,\eta_k}{\sum_{\ell=1}^{K} p(V_i \mid z_\ell;\phi)\,\eta_\ell} \tag{33}$$

$$= \frac{\prod_{j=1}^{m} \sigma_\phi(z_k, V_{ij})\,p(X_{ij} \mid z_k;\theta)\,\eta_k}{\sum_{\ell=1}^{K} \prod_{j=1}^{m} \sigma_\phi(z_\ell, V_{ij})\,p(X_{ij} \mid z_\ell;\theta)\,\eta_\ell}. \tag{34}$$

In the context of LLMs, the quantity $X_{ij}$ is the prompt and the prompts are randomly assigned to annotators, so we would expect no correlation between the preference type of the annotator and the prompt assigned to them. Thus, all prompts are equally likely given the preference type of the annotator. Hence, we make the following assumption: [Un-correlated Contexts and Latent Preference Types] For all $k, \ell \in [K]$:

$$p(X_{ij} \mid Z_i = z_k;\theta) = p(X_{ij} \mid Z_i = z_\ell;\theta) =: \rho(X_{ij}) \tag{35}$$

Based on this assumption, we can then write:

$$p(z_k \mid V_i;\theta) = \frac{\prod_{j=1}^{m} \sigma_\phi(z_k, V_{ij})\rho(X_{ij})\,\eta_k}{\sum_{\ell=1}^{K} \prod_{j=1}^{m} \sigma_\phi(z_\ell, V_{ij})\rho(X_{ij})\,\eta_\ell} \tag{36}$$

Note that we can write:

$$\sum_{\ell=1}^{K}\prod_{j=1}^{m} \sigma_\phi(z_\ell, V_{ij})\,\rho(X_{ij})\,\eta_\ell = \sum_{\ell=1}^{K}\prod_{j=1}^{m} \rho(X_{ij}) \cdot \prod_{j=1}^{m} \sigma_\phi(z_\ell, V_{ij})\eta_\ell \tag{37}$$

$$= \prod_{j=1}^{m} \rho(X_{ij}) \cdot \sum_{\ell=1}^{K}\prod_{j=1}^{m} \sigma_\phi(z_\ell, V_{ij})\eta_\ell \tag{38}$$

Thus, the terms $\prod_{j=1}^{m} \rho(X_j)$ cancel from the numerator and denominator in Equation equation 36, leading to the simplified formula that is independent of $\pi$:

$$p(z_k \mid V_i;\theta) = \frac{\eta_k \prod_{j=1}^{m} \sigma_\phi(z_k, V_j)}{\sum_{\ell=1}^{K} \eta_\ell \prod_{j=1}^{m} \sigma_\phi(z_\ell, V_j)} \tag{39}$$

## D  $M$-STEP DERIVATION

We aim to solve the following two optimization problems:

$$\eta_{t+1} = \arg\max_{\eta} \mathbb{E}_{Z \sim p(\cdot|V,\theta_t)} \left[ \sum_{i=1}^{n} \log \left( \sum_{k=1}^{K} \eta_k 1\{Z_i = z_k\} \right) \right]$$

$$\phi_{t+1} = \arg\max_{\phi,\rho} \mathbb{E}_{Z \sim p(\cdot|V,\theta_t)} \left[ \sum_{i=1}^{n} \log(p(V_i \mid Z_i; \phi, \rho)) \right] \tag{40}$$

The first optimization problem in Equation equation 40 admits a closed-form solution. Letting $w_{k,t} = \sum_{i=1}^{n} p(z_k \mid V_i; \theta_t)$

$$\mathbb{E}_{Z \sim p(\cdot|V,\theta_t)} \left[ \sum_{i=1}^{n} \log \left( \sum_{k=1}^{K} \eta_k 1\{Z_i = z_k\} \right) \right] = \sum_{i=1}^{n} \sum_{k=1}^{K} p(z_k \mid V_i; \theta_t) \log(\eta_k) = \sum_{k=1}^{K} w_{k,t} \log(\eta_k) \tag{41}$$

Thus the optimization problem that determines $\eta_{t+1}$ takes the simple form $\max_{\eta \in \Delta(K)} \sum_{k=1}^{K} w_{k,t} \log(\eta_k)$. The Lagrangian of this problem is $L(\eta, w_t, \lambda) = \sum_{k=1}^{K} w_{k,t} \log(\eta_k) + \lambda^T(\eta - 1)$. The KKT condition is:

$$\frac{w_{k,t}}{\eta_{k,t+1}} = \lambda \implies \eta_{k,t+1} \propto w_{k,t} \implies \eta_{k,t+1} = \frac{w_{k,t}}{\sum_k w_{k,t}} \tag{42}$$

Moreover, since $\sum_k p(z_k \mid V_i; \theta_t) = 1$, we have $\sum_k w_{k,t} = n$. Thus, the above simplifies to:

$$\eta_{k,t+1} = \frac{1}{n} w_{k,t} = \frac{1}{n} \sum_{i=1}^{n} p(z_k \mid V_i; \theta_t) \tag{43}$$

For the second optimization problem in Equation equation 40, we further decompose the objective:

$$\log(p(V_i \mid Z_i; \phi, \rho)) = \sum_{j=1}^{m} \log(p(V_{ij} \mid Z_i; \phi, \rho)) = \sum_{j=1}^{m} \log(\sigma_\phi(Z_i, V_{ij})) + p(X_{ij} \mid Z_i; \rho) \tag{44}$$

Assuming that the parameter $\rho$ that determines that $p(X \mid Z; \rho)$, according to Assumption C is not subject to joint constraints with the parameter $\phi$, we can drop the second part in the objective, when optimizing for $\phi$:

$$\phi_{t+1} = \arg\max_{\phi} \mathbb{E}_{Z \sim p(\cdot|V,\theta_t)} \left[ \sum_{i=1}^{n} \sum_{j=1}^{m} \log(\sigma_\phi(Z_i, V_{ij})) \right] = \sum_{i=1}^{n} \mathbb{E}_{Z_i \sim p(\cdot|V_i;\theta_t)} \left[ \sum_{j=1}^{m} \log(\sigma_\phi(Z_i, V_{ij})) \right] \tag{45}$$

Moreover, since $\rho$ does not enter in the update rules for $\eta, \phi$, nor in the calculation of the posterior, we can ignore it in our EM-DPO algorithm.

## E  MIN-MAX REGRET OBJECTIVE DERIVATION

We can write, by linearity of expectation:

$$R_z(\pi) - R_z(\pi_z^*) = \mathbb{E}_{x \sim \mathcal{D}, y \sim \pi_z^*(\cdot|x), y' \sim \pi(\cdot|x)} \left[ r_z^*(y, x) - r_z^*(y', x) \right] \tag{46}$$

$$= \beta \mathbb{E}_{x \sim \mathcal{D}, y \sim \pi_z^*(\cdot|x), y' \sim \pi(\cdot|x)} \left[ \log \left( \frac{\pi_z^*(y|x)}{\pi_{\text{SFT}}(y|x)} \right) - \log \left( \frac{\pi_z^*(y'|x)}{\pi_{\text{SFT}}(y'|x)} \right) \right] \tag{47}$$

$$= \beta \left( \mathbb{E}_{x \sim \mathcal{D}, y \sim \pi_z^*(\cdot|x)} \left[ \log \left( \frac{\pi_z^*(y|x)}{\pi_{\text{SFT}}(y|x)} \right) \right] - \mathbb{E}_{x \sim \mathcal{D}, y' \sim \pi(\cdot|x)} \left[ \log \left( \frac{\pi_z^*(y'|x)}{\pi_{\text{SFT}}(y'|x)} \right) \right] \right). \tag{48}$$

For any $z, z' \in \{z_1, \ldots, z_k\}$, we will let:

$$\mathcal{L}_{z,z'} := \mathbb{E}_{x \sim \mathcal{D}, y \sim \pi_{z'}^*(\cdot|x)} \left[ \log \left( \frac{\pi_z^*(y|x)}{\pi_{\text{SFT}}(y|x)} \right) \right] \tag{49}$$

Given the policy parameters we estimated in the EM-DPO section, these quantities can be calculated as simple empirical averages over the annotated data. Moreover, note that since our policy $\pi \in \Pi$ is a mixture policy over the policies $\pi_{z'}^*$, for $z' \in \{z_1, \ldots, z_k\}$ with weights $w \in \Delta(K)$, we can write:

$$R_z(\pi) - R_z(\pi_z^*) = \beta \left( \mathcal{L}_{z,z} - \sum_{k=1}^{K} w_k \cdot \mathcal{L}_{z,z_k} \right) = \beta \sum_{k=1}^{K} w_k \cdot (\mathcal{L}_{z,z} - \mathcal{L}_{z,z_k}), \tag{50}$$

Thus, our minimax regret objective can be simply written as:

$$\min_{w \in \Delta(K)} \max_{z \in \{z_1, \ldots, z_k\}} \left[ \sum_{k=1}^{K} w_k \cdot (\mathcal{L}_{z,z} - \mathcal{L}_{z,z_k}) \right]^+ = \min_{w \in \Delta(K)} \max_{z \in \{z_1, \ldots, z_k\}} \max \left\{ 0, \sum_{k=1}^{K} w_k \cdot (\mathcal{L}_{z,z} - \mathcal{L}_{z,z_k}) \right\} \tag{51}$$

Introducing a fake preference population $z_0$ that always has 0 regret, i.e. $\mathcal{L}_{z_0,z_0} = \mathcal{L}_{z_0,z_k} = 0$, we can re-write the above objective simply as:

$$\min_{w \in \Delta(K)} \max_{z \in \{z_0, z_1, \ldots, z_k\}} \sum_{k=1}^{K} w_k \cdot (\mathcal{L}_{z,z} - \mathcal{L}_{z,z_k}) \tag{52}$$

# F  ADDITIONAL EXPERIMENT DETAILS

## F.1  IMDB DATA GENERATION

We use the IMDb dataset Maas et al. (2011) to generate a synthetic preference dataset. More specifically, we use a publicly available adaptation of the IMDb dataset[1]. This dataset uses the first 20 tokens from the original IMDb datasetMaas et al. (2011) as the prompt and then two responses are generated for each prompt using a GPT-2 Large model that is fine-tuned on the IMDb dataset. We synthetically generate preference data for two user groups using this dataset. We select a random subset of 5,000 prompts and assign it equally among 5 users (1,000 preferences per user). This constitutes the first sub-group and these users always prefer the response that is shorter in length. We automatically label the preference data for this group by simply counting the number of words in the response. Next, we select 15,000 prompts from the remaining data and assign it to 15 users (1,000 preference per user). This is the second sub-group where the users always prefer the response that is more grammatically correct. We use LanguageTool[2] to automatically to find the number of grammatical errors in a given text and divide this number by the length of the text to get a correctness score. The users prefer the response with a higher correctness score.

## F.2  CLUSTER-DPO

The Cluster-DPO policy is generated as follows. We naively cluster the 20 users into 2 user subgroups using $k$-means clustering on the average embedding of all the preferred texts of that user. Embeddings are generated using the RoBERTa-Large model Liu (2019). Then, we train a DPO policy on each cluster separately and combine them using Algorithm 2; we are essentially replacing the EM-DPO step with a naive clustering step.

## F.3  HYPERPARAMETERS

Table 1 shows the hyperparameters for the bandit experiment and Table 2 for the LLM experiment. We ran the bandit experiment on one A100 GPU. On average, the code took approximately 1 hour to run. The LLM experiment was run on 5x NVIDIA A6000 GPUs. Every step of the EM algorithm took about 40 minutes to run for a grand total of 13 hours.

---

[1]Modified IMDb dataset

[2]LanguageTool

| Hyperparameter | Value |
|---|---|
| Neural Network Layers | 3 |
| Neural Network Hidden Dimension | 10 |
| Learning Rate | 0.01 |
| Optimizer | Adam |
| DPO Regularization Constant Beta | 1 |
| Max Epochs for Optimization | 1000 |
| Max Steps for EM-DPO | 100 |
| Max Steps for MINMAX-DPO | 1000 |
| Seed (numpy and torch) | 123 |

Table 1: Hyper-parameters for the contextual bandit experiment

| Parameter | Value |
|---|---|
| Learning Rate | 5e-7 |
| Beta | 0.1 |
| Max Text Length (Prompt + Response) | 512 |
| No. of Training Epochs | 1 |
| No. of Evaluation Examples | 256 |
| Optimizer | RMSprop |
| No. of Warmup Steps for Learning Rate | 150 |
| No. of Iterations of the EM Algorithm | 10 |
| No. of Prompts to Estimate the Regret Matrix | 512 |
| Eta for Algorithm 2 | 0.05 |
| Total Steps for Algorithm 2 | 10000 |
| No. of Examples for Evaluation | 256 |
| Seed (DPO), Seed1 (Evaluation), Seed2 (Evaluation) | 0, 42, 62 |

Table 2: Hyper-parameters for the IMDb LLM experiment

