# OpenReview forum: "Direct Preference Optimization With Unobserved Preference Heterogeneity"
_ICLR.cc/2025/Conference — ICLR 2025 Conference Withdrawn Submission_

### Official Review · Reviewer_pTfi · 2024-10-19

**Soundness:** 1
**Presentation:** 1
**Contribution:** 2
**Rating:** 3
**Confidence:** 4

**Summary:**

This work considers the problem of LLM alignment given a dataset with differing preferences from multiple annotators. The authors propose two algorithms, firstly EM-DPO which learns a space of models across the latent preference types of annotators, and secondly MinMax-DPO an algorithm which produces a single policy from the space of models that is robust to the preferences of different groups. The authors conduct a synthetic experiment and an LLM experiment to empirically validate the performance of these algorithms.

**Strengths:**

Well motivated: The motivation of the work is clear and easy to follow. Alignment approaches that are aware of differing preferences is an important problem. The paper clearly explains why this problem is important motivates their solution well.

Originality: Whilst the idea of clustering and training robust models in the RLHF alignment setting is not novel [1], learning the policies directly via direct preference optimization is a novel and interesting approach.

[1] Souradip Chakraborty, Jiahao Qiu, Hui Yuan, Alec Koppel, Furong Huang, Dinesh Manocha, Amrit Singh Bedi, and Mengdi Wang. Maxmin-rlhf: Towards equitable alignment of large language models with diverse human preferences. arXiv preprint arXiv:2402.08925, 2024.

**Weaknesses:**

In my opinion the paper has several weaknesses that lead to me recommending to reject this work in its current state.

Quality and clarity of the writing: The quality of the writing from Section 4 onwards means that the details of the paper are hard to parse and require significant time and effort to properly understand. For example:

-	In Section 4 & 5 all the variables should be defined in a clearer manner e.g. the clearest definition of j is found at the top of page 6, a page after it has been introduced in notation, the range of values the latent factors take should be introduced alongside the latent factors, and consistent notation established for important variables.

-	The introduction of ideas should be re-ordered and clearly explain e.g. equation 6 should be introduced as the M-step only after the clear definition of the latent variables and likelihood, independence assumptions introduced clearly in the text.

-	Algorithms should be fixed with clear line numbers and figures included in the paper should not be screen shots.

Lack of Empirical Investigation: The empirical aspect of this work consists of one synthetic bandit experiment and an LLM experiment run with one model on one dataset.

- Lack of error bars: Many results lack error bars e.g. Figure 2 and 4, from runs across multiple seeds. This is essential to support the statistical significance of the results.

- Lack of real world datasets: Whilst the results on the IMDB dataset seem promising, real world datasets with known annotator labels or a variety of preferences [2,3] should be used to further support the applicability of the work to real world problems.

-  Empirical analysis of EM-DPO: In experiments run with real world data the quality of the clusters learnt should be examined and analysed. Do the clusters found with EM-DPO align with the known annotator labels or differing preferences in the dataset? How does the number of clusters K affect the quality of the robust method? These might be interesting questions to consider empirically.

Missing References: Please include other robust alignment approaches such as [1] in the related work.

Reproducibility Statement: To make the results simple to reproduce please use a service such as Anonymous Github to include anonymised code in your repo. I appreciate this may not be possible in certain cases.

To increase my score the authors must re-write the paper from Section 4 onwards ensuring the ideas are clearly presented, variables defined, algorithms bug free, and graphs cleanly generated. The authors should run experiments across multiple seeds, introduce results on two other pluralistic alignment datasets, and empirically analyse the EM-DPO algorithm in the LLM setting.

[1] Shyam Sundhar Ramesh, Yifan Hu, Iason Chaimalas, Viraj Mehta, Pier Giuseppe Sessa, Haitham Bou Ammar, and Ilija Bogunovic. Group robust preference optimization in reward-free rlhf. arXiv preprint arXiv:2405.20304, 2024.

[2] Esin Durmus, Karina Nyugen, Thomas I Liao, Nicholas Schiefer, Amanda Askell, Anton Bakhtin, Carol Chen, Zac Hatfield-Dodds, Danny Hernandez, Nicholas Joseph, et al. Towards measuring the representation of subjective global opinions in language models. arXiv preprint arXiv:2306.16388, 2023.

[3] Hannah Rose Kirk, Alexander Whitefield, Paul Röttger, Andrew Bean, Katerina Margatina, Juan Ciro, Rafael Mosquera, Max Bartolo, Adina Williams, He He, et al. The prism alignment project: What participatory, representative and individualised human feedback reveals about the subjective and multicultural alignment of large language models. arXiv preprint arXiv:2404.16019, 2024.

**Questions:**

The authors raise interesting points in the discussion and limitations section around how their assumption that the annotator and prompt are uncorrelated. If the two are correlated, how might this affect the learnt clusters in this setting? Preference labels can be collected from users, who often write the prompts they proceed to label, this might be another interesting case where this assumption is violated.

How do the learnt clusters correspond to the attributes of annotators in practice?

---

### Official Review · Reviewer_Eog3 · 2024-10-29

**Soundness:** 2
**Presentation:** 3
**Contribution:** 2
**Rating:** 5
**Confidence:** 4

**Summary:**

This paper proposes an innovative approach to address the challenge of preference heterogeneity in reinforcement learning from human feedback (RLHF) frameworks. Unlike traditional RLHF, which typically assumes uniform preferences, this paper introduces a model that accommodates diverse annotator preferences through a two-stage algorithm. First, an Expectation-Maximization Direct Preference Optimization (EM-DPO) algorithm clusters annotators into latent preference groups. Then, a MinMax-DPO algorithm combines these group-specific policies into an equitable single policy that minimizes the worst-case regret across preference groups. The experimental results demonstrate that this approach provides a more socially equitable policy than conventional DPO, making it a promising alternative to traditional RLHF.

**Strengths:**

* Originality: The paper innovatively addresses preference heterogeneity, a previously overlooked aspect in RLHF. By incorporating an expectation-maximization framework and regret-based ensemble learning, the authors propose an approach that is both unique and impactful.

* Clarity: The paper is mostly clear and well-structured, making it accessible to researchers familiar with RLHF and DPO. Figures and diagrams enhance understanding, although some mathematical sections could be streamlined for accessibility.

**Weaknesses:**

* Assumptions on Preference Homogeneity within Clusters: The model assumes that subgroups are internally homogeneous, which might oversimplify the diversity within real-world preference groups. A discussion on the implications of this assumption and potential strategies for mitigating intra-group variance could be beneficial.

* Limited Real-World Validation: While the experiments are rigorous, real-world datasets could better test the model’s scalability and robustness under diverse real-world conditions.

* Unrealistic problem setting: given limited human feedback data, the per-prompt response pair is usually labeled by very few raters. And the raters have the population bias. How do authors verify that the raters represent a good population distribution among all human in the world?

* Hyperparameter selection: the paper fails to discuss a rigorous way to select number of hidden groups. The number of groups can be even prompt or topic dependent. How to verify that the groups can be differentiated as clusters instead of a continuous spectrum?

* Specific to DPO: this approach should not be specific to DPO, it should be more generalized and the latent heterogeneity can be addressed in reward modeling.

**Questions:**

See weaknesses

---

### Official Review · Reviewer_iWgU · 2024-11-03

**Soundness:** 3
**Presentation:** 2
**Contribution:** 3
**Rating:** 5
**Confidence:** 4

**Summary:**

The paper proposes an additional approach to Direct Preference Optimization (DPO) for language model alignment that addresses unobserved heterogeneity in human preferences. Traditional DPO methods optimize language model alignment by directly using preference data but assume a uniform preference model. This paper presents two new methods, EM-DPO and MinMax-DPO, which allow for diverse preferences among annotators by identifying latent preference types. The EM-DPO algorithm assigns soft clusters to annotators and optimizes policies for each preference type. The MinMax-DPO approach then combines these policies into an equitable ensemble that minimizes worst-case regret across different annotator subgroups. Experiments demonstrate the efficacy of these methods in achieving more fair alignment policies, particularly for underrepresented preference groups.

**Strengths:**

1. The paper provides clear explanations for the motivation and design of both algorithms.
2. Using regret minimization for fair representation across preference groups is nice.
3. The proposed algorithms may be suitable to many of the current problems in llm alignments, such as gender, occupation biases, etc.

**Weaknesses:**

1. Despite the approach is interesting, the writing is rather poor. I would recommend the authors to polish the writing as well as fix simple latex usage, for example, \citet and \citep are different.
2. The experiment section is not convincing enough, I was expecting more practical experiments on more interesting tasks using more recent models, such as llama 3 1b, 3b, 8b or other similar more modern models, but using gpt-2 feels a bit outdated for this approach.

**Questions:**

I don't have specific questions.

---

### Official Review · Reviewer_HoN1 · 2024-11-05

**Soundness:** 2
**Presentation:** 2
**Contribution:** 3
**Rating:** 3
**Confidence:** 3

**Summary:**

The authors propose a two step Preference Optimization algorithm to learn a model aligned to human preferences under the assumption of heterogeneity between the rater demographics.

Their approach involves first learning K policies each optimized with an importance weighted DPO objective, where the importance weights are the posterior probability of an individual (prompt, preference) pair belonging to one of K latent subgroups.

Once these subgroups are identified, the authors propose to use a min-max objective that minimizes the regret of a policy over the worst possible assignment of the groups to each latent variable.

**Strengths:**

1) Modeling the heterogeneity of rater populations when aligning LLMs is an important and challenging research question. The authors propose the use of latent variable approaches towards this problem which seems compelling.

2) The authors further propose a framework that not only learns policies conditional on patient latent demographics, but further propose a minimax style objective that is sound and seems well motivated.

**Weaknesses:**

1) EM Style algorithms are brittle and can lead to quick degenerate results if not subjected to proper regularization. In the paper the authors do not seem to enforce any constraints/regularization on the posterior counts to ensure identifiability. This makes me skeptical of the soundness of this framework.


2) While the ideas and approach presented in the paper are interesting, the execution and experimental section is less convincing. For instance, the presented experiments involve only two rater subpopulations that were synthetically generated. Identifiability with 2 subgroups holds trivially. For a realistic experimental set-up it would be expected to have a real-world/synthetic dataset of atleast 3 or more rater subpopulations.

3) The presentation of the paper needs substantial improvement. For instance the figures 2 & 3 lack sufficient detail/clarity. Notation is incomplete. What is |H| in Algorithm 1? Figure 4 is hard to parse as a bar graph, perhaps better presented as a pareto curve?

**Questions:**

1) EM style algorithms result in soft assignments P(Z) to the K latent clusters. As an alternative to the the min-max objective, a min over the Expected risk where the expectation is computed over the soft assignments might be another compelling metric to compare with as a baseline. Have the authors considered this?

2) In (18) the authors say they only use the positive part of regret? Why? Can the authors please shed more light?

---

### Note · Authors · 2024-12-04

**Comment:**

We appreciate the reviewers' valuable feedback and will use it to design a more comprehensive set of experiments to enhance the paper.

**Withdrawal Confirmation:**

I have read and agree with the venue's withdrawal policy on behalf of myself and my co-authors.